# Espresso: Efficient Forward Propagation for Binary Deep Neural Networks

**Fabrizio Pedersoli**
University of Victoria
fpeder@uvic.ca

**George Tzanetakis**
University of Victoria
gtzan@uvic.ca

**Andrea Tagliasacchi**
University of Victoria
ataiya@uvic.ca

## Abstract

There are many applications scenarios for which the computational performance and memory footprint of the prediction phase of Deep Neural Networks (DNNs) need to be optimized. Binary Deep Neural Networks (BDNNs) have been shown to be an effective way of achieving this objective. In this paper, we show how Convolutional Neural Networks (CNNs) can be implemented using binary representations. *Espresso* is a compact, yet powerful library written in C/CUDA that features all the functionalities required for the forward propagation of CNNs, in a binary file less than 400KB, without any external dependencies. Although it is mainly designed to take advantage of massive GPU parallelism, *Espresso* also provides an equivalent CPU implementation for CNNs. *Espresso* provides special convolutional and dense layers for BCNNs, leveraging *bit-packing* and *bit-wise* computations for efficient execution. These techniques provide a speed-up of matrix-multiplication routines, and at the same time, reduce memory usage when storing parameters and activations. We experimentally show that *Espresso* is significantly faster than existing implementations of optimized binary neural networks ($\approx 2$ orders of magnitude). *Espresso* is released under the Apache 2.0 license and is available at http://github.com/fpeder/espresso.

## 1 Introduction

Convolutional Neural Networks have revolutionized computer vision, pushing the task of object recognition beyond human capabilities (Krizhevsky et al., 2012; Simonyan & Zisserman, 2014; Szegedy et al., 2015). Deep Neural Networks (DNN), have also been successfully applied in other fields, such as speech recognition (Graves et al., 2013; Hinton et al., 2012) and automated translation (Bahdanau et al., 2014; Sutskever et al., 2014). Despite achieving impressive classification accuracy results, DNNs require too much memory and power to be used effectively on embedded or low-power devices. Many networks consume a considerable amount of memory. Memory remains a very limited resource on mobile platforms making harder the usage of trained DNNs [1]. Even when memory is not an issue, DNNs remain very computationally intensive, and can quickly drain the battery. Reducing the computational load does not only improve energy efficiency, but can also enable further applications. For example, when processing real-time object classification on mobile, being able to perform faster predictions frees up computational resources that can be spent on tasks such as speech recognition and analysis. Therefore, there is a substantial interest in reducing the computational and memory requirements of DNNs.

**Efficient deep neural networks** One way to achieve this target is to use specialized hardware for DNNs. Another strategy is to reduce the network's memory footprint and associated computation, hence increasing its efficiency. Such solutions are preferable as they can be implemented in software without requiring specialized hardware. In our research we follow the software approach, and focus our attention to *quantized* networks. In this case, the parameters are stored as "small" integers

---

[1]for example, the popular AlexNet (Krizhevsky et al., 2012) and VGG (Simonyan & Zisserman, 2014) architectures consume respectively $\approx 250\,\mathrm{MB}$ and $\approx 520\,\mathrm{MB}$

(typically less than 8-bit) instead of single precision floating point numbers (32-bit). In particular, we consider the *binary* deep neural networks (BDNN) proposed by Hubara et al. (2016) where parameters and activations are 1-bit integers: $\{-1, +1\}$. At the expense of a relatively small decrease in accuracy, BDNNs can considerably reduce memory usage, and result in faster execution time (i.e. forward propagation). Further, note that potential hardware implementation of BDNNs would also be cheaper due to the reduced number of required FPUs. While these results are highly promising, currently only *proof-of-concept* implementations of BinaryNets have been published (Hubara et al., 2016). Therefore, the availability of a flexible end-to-end framework, with particular emphasis placed on computational efficiency, can enable further research on BDNNs, as well as its application to practical scenarios.

**Contributions**   With *Espresso* we provide an optimized framework for BDNNs capable of achieving state-of-the-art run-time *performance* with minimal *memory* footprint while being numerical equivalent to their non-optimized binary counterpart. *Espresso* provides a complete optimized framework for BDNNs supporting both the *dense* and the *convolutional* layer. Current state-of-the-art optimized BDNNs implementations are limited to the fully connected layer, with the serious drawback of not being able to run optimized state-of-art convolutional BDNNs (BCNNs). While our work is a necessary stepping stone towards optimization of training routines, in this paper we focus on the optimization of forward-propagation (i.e. testing), rather than back-propagation (i.e. training). *Espresso* is designed to have no external dependencies. This not only results in a highly optimized implementation of BDNNs, but also substantially simplifies its deployment in practical applications, such as those executing on mobile or embedded devices.

## 2   RELATED WORK

Improving the performance of DNNs can be achieved at either the hardware or software level. At the hardware level, chipsets that are dedicated to DNN execution can outperform general-purpose CPUs/GPUs (Jouppi, 2017; Han et al., 2016). At software level one approach is to design simpler architectures, in terms of overall floating point operations, that can offer the same accuracy as the original model (Iandola et al., 2016). Another approach is to prune the weights (Guo et al., 2016), or even entire filters (Li et al., 2016), that have low impact on the activations such that a simpler model can be derived. These simplified models can be further compressed by weight sharing (Han et al., 2015). Finally instead of removing connections, another approach is to *quantize* the network weights (Courbariaux et al., 2014) such that computations can be executed more efficiently.

**Quantized networks**   In quantized networks, the objective is to train DNNs whose (quantized) weights do not significantly impact the network's classification accuracy. For example, Courbariaux et al. (2014) show that 10-bits are enough for Maxout Networks, and how more efficient multiplications can be performed with fixed-point arithmetic. Continuing this trend, more aggressive quantization schemes, up to ternary (Zhu et al., 2016), have also been studied.

**Binary Deep Neural Networks (BDNN)**   Recently, Courbariaux et al. (2015) showed that a network with *binary* $\{-1, +1\}$ weights can achieve near state-of-the-art results on several standard datasets. Binary DNNs (BDNNs) were shown to perform effectively on datasets with relatively small images, such as the permutation-invariant MNIST (LeCun et al., 1998), CIFAR-10 (Krizhevsky et al., 2009) and SVHN (Netzer et al., 2011). Recently, Rastegari et al. (2016) show that binarized CNNs can perform well even on massive datasets such as ImageNet (Deng et al., 2009) using binarized versions of well-known DNN architectures such as AlexNet (Krizhevsky et al., 2012), ResNet-18 (He et al., 2016), and GoogLenet (Szegedy et al., 2015). Similarly interesting results can be achieved by binarizing both DNN weights and activations as showed by Hubara et al. (2016). In this work, the authors introduce *BinaryNet*, a technique to effectively train DNNs where both weights and activations are constrained to $\{-1, +1\}$. *BinaryNet* achieves nearly state-of-the-art accuracy for MLP training on MNIST and CNN training on CIFAR-10. The authors also propose a binary optimized implementation of matrix multiplication which result in $7\times$ faster performance than the base-line non optimized implementation, and, almost $2\times$ faster than Theano (Bergstra et al., 2010). Their core contributions, namely to replace Floating-point Multiply and Add operations (FMAs) with *XNORs* and *bit-counts*, represent the cornerstone over which we build our research.

## 3 THE ESPRESSO FRAMEWORK

*Espresso* provides the user with the necessary tools for executing forward-propagation of DNNs, with particular emphasis placed on convolutional neural networks due to their ubiquitousness in computer vision applications. As the complexity of these networks is cubic to the size of the problem, they are less memory efficient and more computationally intensive than traditional machine-learning algorithms. Identifying the memory and computational bottlenecks of DNNs is therefore essential to enable their practical application. In particular, our primary focus is *GPU-optimized* BDNN architectures, which we refer to as $GPU^{opt}$, but we also support the equivalent floating-point counterparts on heterogeneous architectures, which we refer to as *CPU* and *GPU*. The *CPU* and *GPU* implementations of *Espresso* do not feature binary optimizations because the data is encoded as single precision floating point numbers. However they still utilize an optimized library for matrix multiplication.

**Hybrid DNNs** The *Espresso*'s implementations of tensors and layers come in three variants $\{CPU, GPU, GPU^{opt}\}$. A CPU-tensor is allocated in CPU memory, and is processed on the CPU using sequential code. A GPU-tensor is allocated on GPU main memory and is processed by CUDA kernels. *Espresso* provides functions for *converting* tensors and layers from one variant to the other, and different variants can also be interconnected with each other. Consequently, *Espresso* enables the design of hybrid DNNs consisting of a combination of $\{CPU, GPU, GPU^{opt}\}$ layers.

**The computational bottleneck: dot products** Dense linear algebra is at the heart of deep-learning as deep networks can be viewed as a composition of *matrix-matrix*, *matrix-vector* and *elementwise matrix-matrix or vector-vector* multiplications. The implementation of these dense linear algebra operations relies heavily on the efficient computation of the *dot-product*. The execution of this operator consists of (single precision) *Floating-point Multiply and Add* (FMA) operations. In modern architectures, floating-point multiplications executing on the FPU dominate the complexity of FMAs, and BDNNs address these concerns by replacing FMAs with simpler *bitwise* operations; see Section 4.

**Technical highlights** The superior computational performance of *Espresso* derives from three main technical contributions: (1) the use of bit-packing in network layers, (2) better memory layout and management, and (3) the use of custom optimized CUDA kernels. Through the use of bit-packed layers, *Espresso* can execute a forward operation without the need for expensive memory re-arrangements employed by existing implementations. As dynamic memory allocation on GPUs is a performance bottleneck, *Espresso* implements a custom memory allocator that pre-allocates memory at start-up, and replaces the traditional *malloc* and *free* system calls. Finally, matrix multiplications are performed with CUDA kernels that have been adapted to bit-packing, and only resort to XNORs and bit-counts.

## 4 BINARY DEEP NEURAL NETWORKS (BDNNS)

In this section, we overview the fundamental characteristics of BDNNs (Hubara et al., 2016) that inform the basics of *Espresso*'s design. In BDNNs, computationally intensive FMA operations are replaced by *XNOR* (for multiplications) and *bit-count* (for additions), enabling significant computational speed-ups. In particular, XNOR is a simpler machine instruction compared to floating point multiplication, and therefore achieves much higher throughput on many architectures. More importantly, a single XNOR step can execute multiple 64-bit wide blocks of dot-products, further increasing the overall computational efficiency. In what follows, we describe how a network is binarized, detail a compressed memory layout enabling efficient execution of dot-products, show how to re-interpret input data to allow execution on fixed-precision input (e.g. images), and provide a few notes regarding the training procedure.

### 4.1 NETWORK BINARIZATION

A BDNN is composed of a sequence of $k = 1, \ldots, L$ layers whose weights $W_k^b$ and activations $a_k^b$ are binarized to the values $\{-1, +1\}$. The superscript $b$ in the notation indicates binary quantities.

Weights and activations are $\{-1, +1\}$, but at the hardware level they must be encoded as $\{0, 1\}$. Our convention is to encode $-1 \rightarrow 0$ and $+1 \rightarrow 1$. Amongst many possible choices, e.g. stochastic binarization (Courbariaux et al., 2015), we employ the following activation function due to its efficient implementation:

$$x^b = \text{sign}(x) = \begin{cases} +1 & x \geq 0 \\ -1 & \text{otherwise} \end{cases} \tag{1}$$

## 4.2 BIT-PACKING

The weights of a BDNN can be stored in the bits of a 64-bit word. One immediate advantage of bit-packing is to drastically reduce the memory usage by a $32\times$ factor. An even more significant advantage is the ability to process multiple values at the same time using registers. This is particularly useful for dot-products: with bit-packing we can compute a dot-product of $64$ element vectors by using just one XNOR and one bit-count. Furthermore, modern computer architectures provide a hardware instruction for counting the number of bits set to 1 in a given word. Assuming binary vectors $a, b \in \mathbb{B}^{1 \times N}$ where $N$ is a multiple of $64$, the dot-product is then equivalent to:

$$a \cdot b \equiv N - \left( \sum_{i=1}^{N/64} \text{bitcount}(\text{XNOR}(a_i, b_i)) \right) \ll 1 \triangleq a \odot b \tag{2}$$

where $\ll$ represents the bit-shift operator. This simple computation becomes the building block of optimized BDNNs as binary matrix-matrix or matrix-vector operations are computed in this fashion.

## 4.3 INPUT DATA BINARIZATION

BDNNs require binary input data, which is not typically available at the first layer of the network. However, the input data usually comes in a fixed precision format (e.g. 8-bit/channel in RGB images). Therefore, the optimized computation of dot-products can still be applied if we split the input data according to bit-planes, and then sum back each contribution according to the corresponding weight. For instance, if with $\langle a \rangle_n$ we indicate the $n$-th bit of a fixed precision vector, and with $i$ the corresponding bit-plane, we obtain:

$$a \cdot b \equiv \sum_{i=0}^{n-1} 2^i \langle a \odot b \rangle_i \tag{3}$$

## 4.4 TRAINING

When training a BDNN, it is important to note that the gradient is computed with the binary weights, but is accumulated with floating point precision (Hubara et al., 2016). That is because the optimizer needs sufficient precision to make a reliable update. In addition, the derivative of the sign function, which is zero almost everywhere, cannot be used for back-propagation. To overcome these issues, the *straight-through estimator* (Bengio et al., 2013) is employed, where 1 is back-propagated if the floating point argument $|x| \leq 1$, and 0 otherwise. Finally, during training weights are clipped to $[-1, 1]$ to avoid a large growth of the floating point weights that would not have an impact on the binary weights.

## 5 ESPRESSO ARCHITECTURE

The principal components of our framework are *tensors*, *layers* and the *network*. These components are organized as a hierarchy. Tensors are $n$ dimensional matrices used for storing *inputs*, *weights* and *activations* (outputs). A layer processes an input tensor and produces an output tensor, while a network consists of a concatenation of layers.

## 5.1 TENSORS

In *Espresso*, each element of a tensor $A \in \mathbb{R}^{M \times N \times L}$ is identified by the triplet $m, n, l$, where $m \in [0, M)$ indicates the row, $n \in [0, N)$ indicates the column, and $l \in [0, L)$ indicates the channel.

A tensor is stored in memory using row-major order with interleaved channels. Therefore, according to this layout, the element $A_{m,n,l}$ is found at position $(mN + n)L + l$ in linear memory.

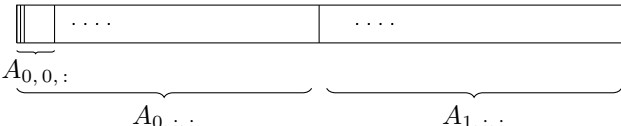

We use the notation $A_{m,n,:}$, to indicate all the channels of the $(m,n)$-th element. Using the same storing scheme *Espresso* also defines bit-packed tensors for *GPU$^{opt}$* implementations but with the following changes to further increase its performance. Bit-packing is performed according to the number of channels: when $L > 1$ bit-packing is done along the $l$ dimension; when $L = 1$ bit-packing is done along the $n$ dimension. For *convolutional* layers this packing direction enables

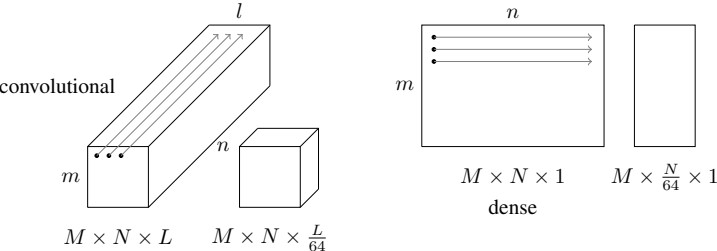

efficient memory access when unrolling/lifting a tensor, which would have not been possible if either $m$ or $n$ had been chosen instead. More specifically, this layout is optimal for retrieving a pixel neighborhood as needed by convolution without requiring the layout to be changed. Further, typically a large number of filters are used resulting in an increase of tensor dimension in the $l$ direction, while the $m$ and $n$ dimensions are progressively shrunk by pooling layers. For other layer types, $n$ is the most efficient packing direction, as neurons are stored along rows and their number decreases as we move toward later stages in the network.

## 5.2 LAYERS

*Espresso* provides the following layer types: *Input*, *Convolutional*, *Pooling*, *Dense* (i.e. fully connected) and *Batch-normalization*. Each layer is characterized by its size, tensor parameters and output. The *Espresso* API defines for each layer a *forward* function that computes the output of a layer given an input tensor, and a function for applying *non-linearity* to the outputs of convolutional and dense layers. Moreover, the convolutional layer features additional functions for *pooling* and *unrolling*.

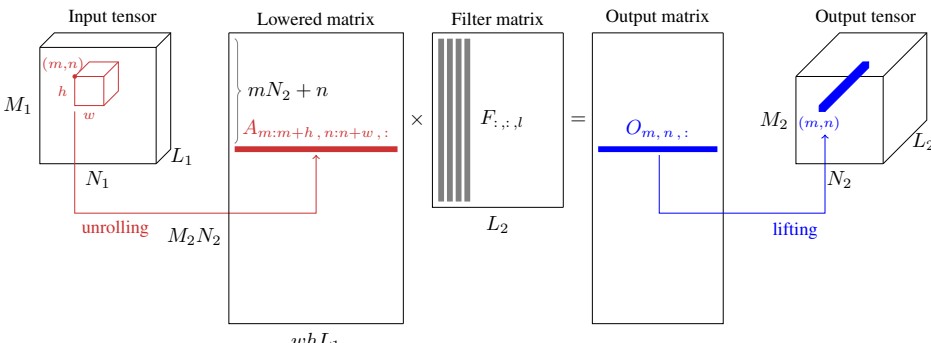

Figure 1: *unrolling* and *lifting* operations for CNN layers

**Convolutional layers**    In our framework, 2D convolutions are computed through matrix multiplications – an operation involving a very high reuse of data. For both *CPU* and *GPU*, this computation is performed by sectioning data in amounts that are cache-friendly (Dongarra et al., 1990), resulting in implementations attaining close to peak computational performance. However, in order to express convolution as matrix multiplication we need to re-organize the input memory appropriately. This is achieved through the *unrolling* procedure; see Figure 1. It consists of transforming a tensor into a matrix where each row is formed by unrolling the tensor data contained in each convolution sliding volume. The unrolled matrix is then multiplied by the filter matrix. Finally, the result of the convolution is reordered back to tensor by using the *lifting* procedure. In *Espresso* we do need to manually lift the convolution result in order to undo the unrolling: thanks to our tensor representation this happens automatically and at zero cost. *Espresso* provides CUDA kernels for the *unrolling* and *pooling* of tensors for both *GPU* and $GPU^{opt}$ implementations.

**Efficient Matrix multiplication**    Matrix-vector multiplications are fundamental operations of both dense and CNN layers. For the *CPU* architecture, we use the OpenBLAS library (Xianyi et al.) to implement these operations. For *GPU* and $GPU^{opt}$ architectures, the CUDA kernels are based on MAGMA(sgemm) (Abdelfattah et al., 2016), modified to make it compatible with our binary data representation. These kernels for matrix multiplication feature *register blocking* optimization: since the introduction of *Fermi* architectures the number of registers have been increased, while register access latency has been substantially reduced compared to shared-memory; hence caching at the register-memory level results in considerably faster throughput (Nath et al., 2010). *Espresso* first fetches the tiles of the two matrices into shared-memory and then process sub-tiles using registers. In the $GPU^{opt}$ variant, we modify the code by replacing blocks of 64 (or blocks of 32 for $GPU^{opt}$ 32) single precision multiply and add (FMA) operations with XNOR and bit-count using packed tensors. We also re-tune the kernel block size parameters for improving the performance on reduced size matrices.

**Zero-padding for convolutions**    Typical CNN implementations apply a tensor convolution in a "same" configuration, where the sizes of input and output tensors matches. This is achieved by zero-padding input tensors, but in convolutional $GPU^{opt}$ layers the *zero-padding* of the input introduces the side-effect of making the data ternary $\{-1, 0, +1\}$. We deal with this problem by treating the data as if it was binary (zero is considered a minus one) and fix the results of the convolution at these corner-cases in post-processing. This allows us to leave the convolution kernel code – the computational bottleneck of the code – untouched. The corner-cases are fixed using a highly efficient kernel which executes an element-wise sum between the results of the convolution and the correction matrix. The correction matrix is computed once, when the $GPU^{opt}$ layer is loaded, and it simply consists of the convolution of the layer's weights with a $(+1)$-padded zero-tensor.

**Converting a network to Espresso**    A DNN in *Espresso* is defined as a combination of layers, which is loaded at run-time by reading its parameters file. The parameters file specifies the storage format of all the layers, as well as their weights. Therefore, it completely specifies a DNN as layers are stored sequentially. Training of the network is done by BinaryNet (Hubara et al., 2016); the resulting parameters are converted to the *Espresso* format by utility script distributed together with our sources.

## 6   EVALUATION

The performance of our framework is evaluated in terms of average computational time needed to perform a particular task. The execution times, averaged over 100 experiments, are obtained on a machine equipped with an NVIDIA GeForce GTX 960 with 2GB of RAM, and a Intel® dual-Xeon® X5660 @ 2.80 GHz. In *CPU* mode, we configure the OpenBLAS library for matrix multiplication to use all the 24 available cores.

**Experimental design**    We perform three quantitative evaluations: (Section 6.1) matrix multiplications of two dense square matrices of size $8192 \times 8192$; (Section 6.2) forward-propagations of a Multi-Layer Perceptron (MLP) trained on the MNIST dataset (LeCun et al., 1998); (Section 6.3) forward-propagations of a Convolutional Neural Network (CNN) trained on the CIFAR-10 dataset (Krizhevsky et al., 2009). By Using the same models and datasets, we compare *Espresso*

with: (1) the author provided optimized implementation of BinaryNet (Courbariaux et al., 2015); (2) the optimized BDNN implemented in the Intel Nervana *neon* framework (NervanaSystems); (3) a self-comparison across $\{CPU, GPU, GPU^{opt}\}$ as no binary-optimized implementations of convolutional layers are publicly available. *Espresso* is numerically equivalent to BinaryNet in terms of classification accuracy. Therefore our evaluation focuses on computation speed.

**Public datasets**  The MNIST dataset (LeCun et al., 1998) consists of 60K instances for training and, 10K instances for testing. Each instance is a $28 \times 28$ grayscale image that depicts digits ranging from 0 to 9. The CIFAR-10 dataset (Krizhevsky et al., 2009), consists of 50K training instances and 10K testing instances of $32 \times 32 \times 3$ color images. Images are subdivided into 10 classes (airplanes, automobiles, birds, cats, deers, dogs, frogs, horses, ships and trucks). Since our interest is to asses the real-time performance of binary optimized DNNs, in those experiment we use a batch-size of one, and measure the averaged forward time for each image of the testing-sets for each dataset.

## 6.1 BINARY DENSE MATRIX MULTIPLICATION

Table 1: Averaged time of binary optimized matrix multiplication.

| BinaryNet | Espresso $GPU^{opt}$ 32-bit | Espresso $GPU^{opt}$ 64-bit |
|---|---|---|
| 88 ms | 16 ms (5.5×) | 11 ms (8×) |

In computing dense matrix multiplication, *Espresso* outperforms BinaryNet by a $\approx 8\times$ factor. Much of the gain can be attributed to our optimized kernels, and the use of register blocking: by fetching bigger data from main memory and shared memory, our kernel increases the bandwidth utilization by decreasing the number of memory fetch instructions. The use of 64-bit packing instead of the 32-bit (such as that of BinaryNet), introduces an additional performance improvement. The 64-bit kernel achieves a memory DRAM throughput of $40\,\mathrm{GB\,s^{-1}}$ for reads and $5\,\mathrm{GB\,s^{-1}}$ for writes, while the 32-bit kernel obtain $29.6\,\mathrm{GB\,s^{-1}}$ for reads and $3.6\,\mathrm{GB\,s^{-1}}$ for writes. This translates into the resulting $\approx 25\%$ speed improvement.

## 6.2 MULTI-LAYER PERCEPTRON ON MNIST

Table 2: Average prediction time of the BMLP.

| BinaryNet | Nervana/Neon | Espresso *CPU* | Espresso *GPU* | Espresso $GPU^{opt}$ |
|---|---|---|---|---|
| 18 ms | 17 ms | 37.4 ms | 3.2 ms (5.6×) | 0.26 ms (68×) |

We evaluate the average classification execution time over the MNIST dataset, where we trained the MLP architecture from (Courbariaux et al., 2016, Sec 2.1) with author-provided sources, and then converted it to *Espresso*'s format. In Table 2, *Espresso* achieves a consistent speed-up of $\approx 68\times$ when compared to BinaryNet. As the Nervana/neon implementation of binary network is a BinaryNet derivative, it is affected by the same drawbacks of BinaryNet, and hence achieves comparable performance. Both alternatives have the additional cost of running CUDA through Python/Theano which may introduce further latency in the process. In Table 2, the evaluation over the three variants of *Espresso* shows the expected outcome, with the $GPU^{opt}$ implementation leading the ranking. Note that we are able to achieve a speedup of $\approx 12\times$ on an NVIDIA GTX 960 ($\approx$ 2.5 TFLOPs), although this device has only roughly four times more throughput than the Xeon X5660 ($\approx 500$ GFLOPs without turbo-boost). Through binary optimization, we are able to further increase the performance to $\approx 15\times$ with respect to the GPU implementation. We attribute our computational gains to (1) the use of *binary-optimized* layers, (2) our use of *optimized kernels* for matrix multiplication and (3) *Espresso*'s ability to perform binary optimization of the first layer.

**Binary optimized layers**  An evident drawback of Binary-Net is the need for binarizing/packing the layer's parameters *every time* a forward method is called. In the case of binary optimized networks, the cost of packing the parameters is closely related to the cost of multiplication itself. There-

fore, the reduction of bit-packing function calls leads to a consistent improvement. This motivates our choice of designing specific layers, where bit-packing is done once during network loading.

**Optimized kernels**    BinaryNet employs two bit-packing kernels: one for row-packing, the other for column-packing. Although BinaryNet's pack-by-rows kernel is slightly slower than ours ($8\%$), the pack-by-columns kernel is significantly slower ($\approx 4\times$) due to non-coalesced accesses to global memory. An additional performance gain of $\approx 15\%$ is achieved by swapping matrix-vector in favour of matrix-matrix multiplication kernels when appropriate (i.e. Dense layers with batch size equal to 1); for this reason, *Espresso* also includes the binary-optimized MAGMA(sgemv) kernel.

**First-layer binary optimization**    Another important advantage offered by *Espresso* is the ability to leverage binary optimization in the *first* layer. Since the first stage of a network processes non-binary data, BinaryNet does not feature binary optimization for this layer. However if the input data is split into its constituent bit-planes, binary optimization can still be applied. In particular, we split the input vector in a matrix of 8 rows, and recombine the result after multiplication by a weighted sum. Our experimental results report an overall $\approx 3\times$ performance boost when comparing the full binary optimized network with one in which the first layer is not binary optimized.

Finally, in terms of memory the *GPU*$^{opt}$ implementation requires $4.57\,\text{MB}$ instead of $140.6\,\text{MB}$ as in the case of non binary optimized implementation, resulting in a saving $\approx 31\times$ amount of memory.

## 6.3    Convolutional Neural Network on CIFAR-10

Table 3: Average prediction time of the BCNN.

| Espresso *CPU* | Espresso *GPU* | Espresso *GPU*$^{opt}$ |
|---|---|---|
| $85.2\,\text{ms}$ | $5.2\,\text{ms}\ (16\times)$ | $1.0\,\text{ms}\ (85\times)$ |

To the best of our knowledge, no BDNN implementation of *binary-optimized* CNN layers is publicly available. Our self-evaluation implements the *VGGNet*-like CNN architecture from Hubara et al. (Hubara et al., 2016, Sec. 2.3), and evaluates it across our three modalities: as expected the *GPU*$^{opt}$ implementation achieves significantly better performance.

**Unrolling and pooling**    Note how the *GPU* implementation offers a slightly better improvement over *CPU* with respect to the MLP test, with an $\approx 16\times$ speed-up. In this experiment, the inherent parallelism of unrolling and pooling, and the GPU higher memory throughput explain the behavior. Gains are marginal as FMA still represents the computational bottleneck.

**Bit-packing**    The *GPU*$^{opt}$ implementation results in a $\approx 5\times$ performance gain with the respect to *GPU*. These gains, to binary optimizations, are slightly smaller than those discussed for MLP in Section 6.2. The output of convolutional layers is significantly larger than those of MLP's dense layers, therefore, the computation of bit-packing sign-activation requires more computational effort.

Finally, in terms of memory the *GPU*$^{opt}$ implementation requires $1.73\,\text{MB}$ instead of $53.54\,\text{MB}$ as in the case of non binary optimized implementation, resulting in a saving $\approx 31\times$ amount of memory.

## 7    Conclusions

In this paper we presented *Espresso*, a highly optimized forward-propagation framework for both traditional DNNs as well as BCNNs, that supports heterogeneous deployment on CPU and GPU. While BinaryNet and Nervana/neon BDNN implementations are limited to MLP networks, our framework also supports the popular CNN while simultaneously outperforming state-of-the-art implementations of MLP networks. *Espresso* is highly-efficient, light-weight and self-contained. Computation on the GPU side is done though specifically designed CUDA kernels, which, combined with a more careful handling of memory allocation and bit-packing, allows us to obtain considerable performance improvements. In future work we would like to add training capabilities, and perform additional performance comparisons on larger standard datasets.

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
