# OpenReview forum: "Espresso: Efficient Forward Propagation for Binary Deep Neural Networks"
_ICLR.cc/2018/Conference — Accept (Poster)_

### Official Review · AnonReviewer2 · 2017-11-15
**A full implementation of binary CNN with code**

**Rating:** 7
**Confidence:** 4

**Review:**

This paper builds on Binary-NET [Hubara et al. 2016] and expands it to CNN architectures. It also provides optimizations that substantially improve the speed of the forward pass: packing layer bits along the channel dimension, pre-allocation of CUDA resources and binary-optimized CUDA kernels for matrix multiplications. The authors compare their framework to BinaryNET and Nervana/Neon and show a 8x speedup for 8092 matrix-matrix multiplication and a 68x speedup for MLP networks. For CNN, they a speedup of 5x is obtained from the GPU to binary-optimizimed-GPU. A gain in memory size of 32x is also achieved by using binary weight and activation during the forward pass.

The main contribution of this paper is an optimized code for Binary CNN. The authors provide the code with permissive licensing. As is often the case with such comparisons, it is hard to disentangle from where exactly come the speedups. The authors should provide a table with actual numbers instead of the hard-to-read bar graphs. Otherwise the paper is well written and relatively clear, although the flow is somewhat unwieldy.

Overall, i think it makes a good contribution to a field that is gaining importance for mobile and embedded applications of deep convnets. I think it is a good fit for a poster.

---

### Official Review · AnonReviewer1 · 2017-11-27
**Review revised**

**Rating:** 6
**Confidence:** 3

**Review:**

The paper presents a library written in C/CUDA that features all the functionalities required for the forward propagation of BCNNs. The library is significantly faster than existing implementations of optimized binary neural networks (≈ 2 orders of magnitude), and will be released on github.

BCNNs have been able to perform well on large-scale datasets with increased speed and decreased  energy consumption, and implementing efficient kernels for them can be very useful for mobile applications. The paper describes three implementations CPU, GPU and GPU_opt, but it is not entirely clear what the differences are and why GPU_opt is faster than GPU implementation.

Are BDNN and BCNN used to mean the same concept? If yes, could you please use only one of them?

The subsection title “Training Espresso” should be changed to “Converting a network to Espresso”, or “Training a network for Espresso”.

What is the main difference between GPU and GPU_opt implementations?

The unrolling and lifting operations are shown in Figure 2. Isn’t accelerating convolution by this method a very well known one which is implemented in many deep learning frameworks for both CPU and GPU?

What is the main contribution that makes the framework here faster than the other compared work? In Figure1, Espresso implementations are compared with other implementations in (a)dense binary matrix multiplication and (b)BMLP and not (c)BCNN. Can others ( BinaryNet
(Hubara et al., 2016) or Intel Nervana/neon (NervanaSystems)) run CNNs?

6.2 MULTI-LAYER PERCEPTRON ON MNIST – FIGURE 1B AND FIGURE 1E. It should be Figure 1d instead of 1e?

All in all, the novelty in this paper is not very clear to me. Is it bit-packing?


UPDATE:
Thank you for the revision and clarifications. I increase my rating to 6.

---

### Official Review · AnonReviewer3 · 2017-11-29
**fast implementation of binary networks**

**Rating:** 7
**Confidence:** 1

**Review:**

The paper presents an implementation strategy (with code link anonymized for review) for fast computations of binary forward inference. The paper makes the approach seem straightforward (clever?) and there has been lots of work on fast inference of quantized, low-bit-width neural networks, but if indeed the implementation is significantly faster than commercial alternatives (e.g. from Intel) then I expect the authors have made a novel and useful contribution.

The paper is written clearly, but I am not an expert in alternative approaches in this area.

---

### Decision · Program_Chairs · 2018-01-29
**ICLR 2018 Conference Acceptance Decision**

**Decision:**

Accept (Poster)

**Comment:**

This paper describes a new library for forward propagation of binary CNNs. R1 for clarification on the contributions and novelty, which the authors provided. They subsequently updated their score. I think that optimized code with permissive licensing (as R2 points out) benefits the community. The paper will benefit those who decide to work with the library.